# DECOUPLING BIDIRECTIONAL GEOMETRIC REPRESENTATIONS OF 4D COST VOLUME VIA 2D CONVOLUTION

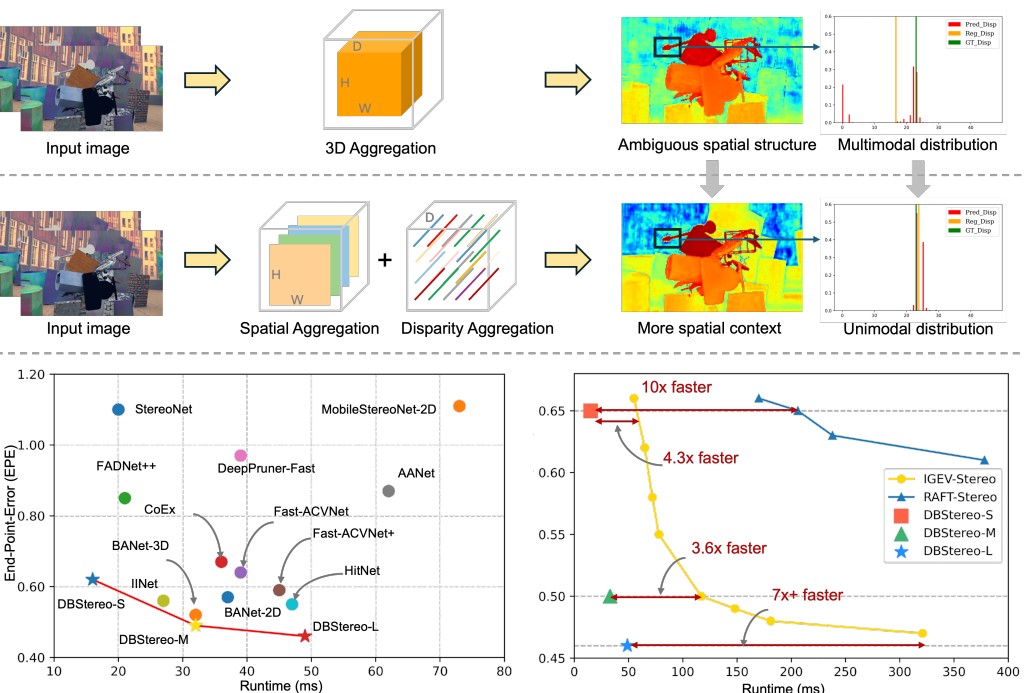

Figure 1: The proposed DBStereo decouple the traditional 3D aggregation into spatial aggregation and disparity aggregation which is merely based on 2D convolutions. The spatial aggregation can incorporate more spatial structure context and the disparity aggregation make the prediction of disparity more concentrated around the ground truth. Our DBStereo outperforms all existing aggregation-based methods (Bangunharcana et al., 2021; Duggal et al., 2019c; Khamis et al., 2018; Li et al., 2024; Shamsafar et al., 2022; Tankovich et al., 2021; Wang et al., 2021; Xu et al., 2022; 2023c; Xu & Zhang, 2020b) in both inference time and accuracy, even surpassing classical iterative-based methods such as RAFT-Stereo (Lipson et al., 2021b) and IGEV-Stereo (Xu et al., 2023b).

## ABSTRACT

High-performance real-time stereo matching methods invariably rely on 3D regularization of the 4D cost volume, which is unfriendly to mobile devices. While methods based on 2D regularization of 3D cost volume struggles in ill-posed regions. In this paper, we propose Decoupling Bidirectional Geometric Representations of 4D cost volume and present a deployment-friendly network DBStereo, which is based on pure 2D convolutions. Specifically, we first provide a thorough analysis of the decoupling characteristics of 4D cost volume. And design a lightweight decoupled bidirectional geometry aggregation block to capture spatial and disparity representation respectively. Through decoupled learning, our approach achieves real-time performance and impressive accuracy simultaneously. Extensive experiments demonstrate that our proposed DBStereo outperforms all existing aggregation-based methods in both inference time and accuracy, even surpassing the iterative-based methods such as RAFT-Stereo and IGEV-Stereo.

Our study breaks the empirical design of using 3D convolution for 4D cost volume and provides a simple yet strong baseline, i.e., the proposed decoupled aggregation paradigm, to facilitate further study.

# 1 INTRODUCTION

Stereo matching has remained a core challenge in computer vision over the past decade, continuously advancing critical applications such as autonomous driving (Yang et al., 2019), industrial robotics (Hsieh & Lin, 2020), and augmented reality (Zenati & Zerhouni, 2007). The essence of the technology lies in establishing accurate pixel-level correspondences between left and right images. However, under the resource-constrained conditions of edge computing devices, simultaneously achieving high matching accuracy and real-time inference remains a significant bottleneck.

With the evolution of deep learning, end-to-end stereo matching frameworks have gradually become mainstream. One of the representative works is PSMNet (Chang & Chen, 2018), which constructs a 4D cost volume and utilizes 3D convolutional network to aggregate it. Such 4D cost aggregation paradigm (Cheng et al., 2024; Duggal et al., 2019a; Liang et al., 2019; Nie et al., 2019; Wu et al., 2019) achieve significant breakthroughs on GPU devices. However, the redundant information inherent in 4D cost volumes force the model to rely on computationally expensive 3D convolutions for regularization, posing substantial difficulties for mobile deployment. In recent years, iterative optimization paradigms (Lipson et al., 2021a; Wang et al., 2024; Xu et al., 2023a; Cheng et al., 2025; Wei et al., 2025), have demonstrated superior performance. Unlike previous aggregation-based methods, these approaches construct 3D correlation cost volumes and progressively refine disparity maps through iterative indexing it, thereby avoiding complex cost aggregation. While reducing computational complexity, the lack of cost aggregation results in cost volumes deficient in global geometric information, leading to disparity discontinuities in occluded regions, mismatches in textureless areas, and artifacts on reflective surfaces. More critically, achieving acceptable accuracy often requires multiple iterations, resulting in inference delays exceeding 100 ms for most methods, which hinders their applicability in real-time scenarios.

Real-time stereo mathcing research (Bangunharcana et al., 2021; Duggal et al., 2019c; Khamis et al., 2018; Li et al., 2024; Shamsafar et al., 2022; Tankovich et al., 2021) can be categorized into two types: 2D CNNs based and 3D CNNs based. Both of them made significant compromises: AANet (Xu & Zhang, 2020a) constructs a 3D correlation cost volume and enhances performance in pathological regions by using deformable convolutions, but its specialized operators pose challenges for deployment on edge devices; MobileStereoNet-2D (Shamsafar et al., 2022) attempts a pure 2D convolutional architecture but suffers severe performance degradation; DeepPruner (Duggal et al., 2019b) narrows the search space by pruning the 4D cost volumes, ACVNet (Xu et al., 2022) filters redundant information via attention weights, yet both still rely on 3D CNNs for aggregation. Empirically, it appears that the informative 4D cost volume can not escape its dependence on 3D CNNs.

In fact, these methods overlook inherent limitations of 3D CNNs in stereo matching: spatial and disparity dimensions share the same receptive fields, while disparity aggregation requires a global receptive field, which leading to degradation; the coupled learning of spatial and disparity features increases the training difficulty. Although FoundationStereo (Wen et al., 2025) recognizes the need for different receptive fields of two dimensions and decomposes a 3D convolution into a spatial 3D convolution and a disparity 3D convolution, it remains a localized refinement of standard 3D convolution rather than addressing the fundamental issue of coupled learning.

In this paper, we propose a novel pure 2D CNN-based framework for 4D cost aggregation that simultaneously achieves real-time performance and high accuracy. We first provide an in-depth analysis of the limitations of 3D regularization networks and introduce our spatial-disparity decoupled aggregation paradigm. Specifically, we first use Disp2Channel operator to transform the 4D cost volume to the 3D one. Then, through our designed Bidirectional Geometry Aggregation (BGA) block consisting of Spatial Aggregation module and Disparity Aggregation module, we decouple the geometric representation of the cost volume into spatial and disparity dimensions. By leveraging 2D CNN-based bidirectional geometric representation decoupling, our method achieves significant improvement. More importantly, our work pioneers a new technical pathway for high-accuracy real-time stereo matching.

Our main contributions are summarized as follows:

- We provide a thorough analysis for the geometric representation of 4D cost volume, breaking away from the traditional coupled aggregation paradigm based on 3D convolutions, and establish a simple yet strong baseline for efficient 4D cost aggregation.

- We design a pure 2D convolutional Bidirectional Geometry Aggregation block to independently capture spatial and disparity representation of the 4D cost volume.

- We demonstrate the effectiveness of our approach, achieving state-of-the-art performance on multiple benchmarks. The proposed decouple aggregation paradigm opens up a new research direction for the community.

## 2 RELATED WORK

Cost aggregation paradigm stereo matching methods (Duggal et al., 2019b; Guo et al., 2019; Kendall et al., 2017; Xu et al., 2022; Xu & Zhang, 2020a) typically follow a four-stage pipeline: feature extraction, cost volume construction, cost aggregation, and disparity regression. Among these, the cost volume serves as the core basis for matching decisions, and its construction quality directly affects final performance.

**Cost volume construction:** Existing cost volume representation can be divided into two categories: the concatenation volume and the correlation volume. GC-Net (Kendall et al., 2017) directly concatenate the features maps of left and right images to construct a 4D concatenation cost volume for all disparities. This dense 4D concatenation volume retains comprehensive information from all channels, and thus exhibit enhanced performance. However, excessive redundant information forces the model to rely on a large amount of 3D convolutions to aggregate and regularize the 4D cost volume, which means high computational and memory cost. RAFT-Stereo (Lipson et al., 2021b) employs the all-pairs correlation constructed based on a similarity matrix derived from left and right image features. However just calculates the feature correlation matrix lacks non-local information and struggling in ill-posed regions. GwcNet (Guo et al., 2019) designed a group-wise correlation cost volume that combines the advantage of these two cost volumes. IGEV-Stereo (Xu et al., 2023b) constructs a geometry encoding volume incorporating context information and local matching clues.

**Cost aggregation:** In order to filter the redundant noise on cost volume, cost aggregation consumes a significant amount of computational resources. DiffuVolume (Zheng et al., 2025) design an effective diffusion-based framework which casts the information filtering as the denoising process of the diffusion model. ACVNet (Xu et al., 2022) proposed the attention mechanism to filter the cost volume and significantly alleviated the burden of cost aggregation. BANet (Xu et al., 2025) utlized spatial attention to separate high-frequency edge regions and low-frequency smooth regions of cost volume. However, these methods still require stacked 3D convolutions to regularize the 4D cost volumes. **Empirically, it seems that high-dimensional cost volume inevitably require dimension-matched convolution to capture the internal correspondences.**

## 3 IS 3D CONVOLUTION NECESSARY FOR 4D COST AGGREGATION?

In learning-based stereo matching, constructing a 4D cost volume ($D \times C \times W \times H$) and applying regularization form the foundation of state-of-the-art paradigms (Chang & Chen, 2018; Guo et al., 2019; Kendall et al., 2017; Xu et al., 2022). A widely adopted convention is to directly employ 3D CNNs to process this 4D tensor. The underlying intuition is powerful and seemingly natural: a high-dimensional tensor appears to inherently require dimension-matched convolution to capture the complex, intertwined relationships across all its dimensions.

However, this empirical design prompts a critical reflection: Is this dimension-matched design truly necessary or optimal? Could it potentially introduce redundancy or even noise? This section delves into the inherent limitations of 3D regularization networks for stereo matching, thereby motivating our novel spatial-disparity decoupled aggregation paradigm based on pure 2D convolutions.

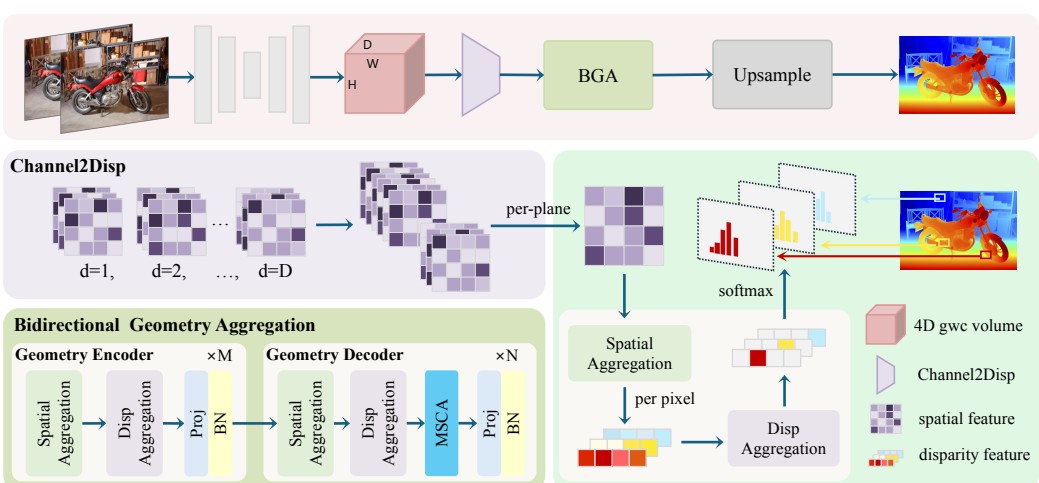

Figure 2: The framework of our proposed DBStereo. The Bidirectional Geometry Aggregation (BGA) block is stacked with multiple Spatial Aggregation modules and Disparity Aggregation modules. The Spatial Aggregation module extracts spatial context for each plane of the cost volume and the Disparity Aggregation modules performs global receptive field disparity filtering on each pixel.

## 3.1 LIMITATIONS OF TRADITIONAL 3D REGULARIZATION NETWORK

In stereo matching based on 4D cost volumes, traditional aggregation paradigms commonly employ 3D convolutions for cost aggregation. Yet, this paradigm suffers from two inherent flaws:

**Coupled Aggregation Pattern:** The traditional paradigm enforces the use of 3D convolution kernels (e.g., $3 \times 3 \times 3$) to extract features from both spatial and disparity dimensions simultaneously. It implies that the model must capture spatial context and disparity context simultaneously within a local region through a single convolution operation. However, this coupled learning approach is susceptible to overfitting due to noise in the training data.

**Slow Receptive Field Expansion in Disparity Dimension:** Due to the coupled aggregation pattern, the expansion of the receptive field in the disparity dimension is inherently tied to that in the spatial dimensions. Constrained by the kernel size, each 3D convolution operation can only increase the receptive field in the disparity direction marginally (e.g., a $3 \times 3 \times 3$ kernel increases it by only 2). To capture sufficient disparity context, a deep stack of 3D CNNs layers is required, directly leading to a dramatic increase in computational cost and memory consumption.

## 3.2 SPATIAL-DISPARITY DECOUPLED AGGREGATION PARADIGM

Based on the above shortcomings, we conducted a thorough analysis of the inherent properties of stereo matching tasks and summarized the following task-specific priors.

**Spatial Local Smoothness Prior:** Adjacent pixels at the same depth possess similar disparity values. This prior is particularly beneficial in low-frequency, textureless regions.

**Disparity Unimodality Prior:** For each single pixel, the correct disparity value should be unique, meaning the disparity probability distribution should be a sharp unimodal distribution.

We propose a novel spatial-disparity decoupled aggregation paradigm that explicitly encodes these two inherent prior into our network architecture, thereby introducing a powerful inductive bias. Specifically, we reshape the high-dimensional 4D cost volume ($D \times C \times W \times H$) into a 3D tensor ($D \cdot C \times W \times H$), decoupling the traditional 4D cost aggregation into two successive pure 2D convolution steps:

**Spatial Aggregation:** A 2D convolution (e.g., $3 \times 3$ kernel) is applied to the spatial dimension of reshaped cost volume ($D \cdot C \times W \times H$). This step focuses on aggregating spatial context within the

same disparity level, effectively smoothing image noise and resolving matching ambiguities in areas like textureless regions.

**Disparity Aggregation:** Traditional 3D CNNs with their limited local receptive field in disparity dimension struggle to capture long-range dependencies, often resulting in a blurred distribution or a multi-modal distribution at edges or a flat distribution in textureless areas. We aim for each aggregation along the disparity dimension to possess a global receptive field and apply 2D CNNs with a $1 \times 1$ kernel to the features after spatial aggregation. It is noteworthy that this $1 \times 1$ convolution essentially performs a global, fully-connected interaction across the entire disparity dimension ($D$), achieving highly efficient optimization of the disparity context. Through this design, our disparity aggregation module enables comparison and competition across the global disparity range, effectively suppressing incorrect disparity responses and facilitating the formation of a more reasonable, sharper unimodal probability distribution, leading to clearer object boundaries.

### 3.3 CONCLUSION

In summary, compared to coupled 3D CNNs, our decoupled structure imposes highly precise inductive biases: spatial aggregation enforces *spatial local smoothness prior*, while disparity aggregation enforces *disparity unimodality prior*. By incorporating these inductive biases into the network architecture, we have significantly reduced the model's search space. The model no longer needs to implicitly learn these fundamental rules from vast amounts of data but instead learns higher level feature representations directly under these inductive biases. This significantly mitigates the risk of the model overfitting to noisy data, thereby achieving a regularization effect far surpassing that of traditional 3D CNNs.

## 4 METHODS

In this section, we introduce the detailed structure of our proposed DBStereo. As shown in Figure 2, unlike previous approaches utilizing 3D CNNs, we decouple the 3D regularization network into a bidirectional geometry aggregation module based on purely 2D CNNs: disparity aggregation module and spatial aggregation module.

### 4.1 FEATURE EXTRACTOR

We employ EfficientnetV2 (Tan & Le, 2021) pretrained on ImageNet (Deng et al., 2009) as our backbone to extract multi-scale feature maps $\{\mathbf{f}_{l,i}, \mathbf{f}_{r,i} \in \mathbb{R}^{C_i \times \frac{H}{i} \times \frac{W}{i}}\}, i = 4, 8, 16, 32$. And a cascade of upsampling blocks are utilized to restore the feature maps to 1/4 resolution of input image. Finally, we obtain multi-scale feature maps $F_{l,i}, F_{r,i} \in \mathbb{R}^{C_i \times \frac{H}{i} \times \frac{W}{i}}$ $i = 4, 8, 16$. Among them, $F_{l,4}, F_{r,4}$ are used to construct the 4D cost volume for subsequent disparity prediction, while $F_{l,4}, F_{l,8}, F_{l,16}$ are utilized to generate spatial attention, further enhancing the robustness of disparity estimation.

### 4.2 COST VOLUME CONSTRUCTION

We construct a 4D group-wise correlation volume (Guo et al., 2019) with features extracted from the left and right images. The left and right features are split into groups and computing correlation maps group by group.

$$\mathbf{C}_{gwc}(d, x, y, g) = \frac{1}{N_c/N_g} \langle \mathbf{f}_l^g(x, y), \mathbf{f}_r^g(x - d, y) \rangle, \tag{1}$$

where $N_c$ denotes the number of feature channels, $N_g$ denotes the number of groups and $d$ denotes the all disparity candidates.

### 4.3 COST AGGREGATION

Given the 4D group-wise cost volume, we first use the Disp2Channel operator to fuse the feature dimension and the disparity dimension of the original cost volume. The core idea of our Disp2Channel transformation is to concatenat the feature maps from all disparity levels, converting the 4D geometric

Table 1: Comparison with the state-of-the-art methods on SceneFlow. Runtime is measured on an RTX 3090 GPU. The `best` and `second best` are marked with colors.

| Paradigm | Method | EPE (px) | D1 (%) | Runtime (ms) |
|---|---|---|---|---|
| Cost aggregation | PSMNet (Chang & Chen, 2018) | 1.09 | 12.1 | 317 |
| | StereoNet (Khamis et al., 2018) | 1.10 | - | 20 |
| | AANet (Xu & Zhang, 2020a) | 0.87 | 9.3 | 93 |
| | AANet+ (Xu & Zhang, 2020a) | 0.72 | 7.4 | 87 |
| | MobileStereoNet-2D (Shamsafar et al., 2022) | 1.11 | - | 73 |
| | FADNet++ (Wang et al., 2021) | 0.85 | - | 21 |
| | CoEx (Bangunharcana et al., 2021) | 0.67 | 4.02 | 36 |
| | Fast-ACVNet (Xu et al., 2023c) | 0.64 | 2.31 | 39 |
| | Fast-ACVNet+ (Xu et al., 2023c) | 0.59 | 2.08 | 45 |
| | IINet (Li et al., 2024) | 0.54 | 2.18 | 26 |
| | BANet-2D (Xu et al., 2025) | 0.57 | 2.50 | 37 |
| | BANet-3D (Xu et al., 2025) | 0.51 | 2.21 | 33 |
| Iterative optimization | RAFT-Stereo (Lipson et al., 2021b) | 0.61 | 2.85 | 380 |
| | IGEV-Stereo (Xu et al., 2023b) | 0.47 | 2.47 | 340 |
| Decoupled aggregation | DBStereo-S (Ours) | 0.63 | 2.36 | 15 |
| | DBStereo-M (Ours) | 0.50 | 1.80 | 33 |
| | DBStereo-L (Ours) | 0.46 | 1.57 | 49 |

representation into a dense 3D representation without altering spatial structure. Specifically, we reshape the volume as follows:

$$C_{3D} = \text{Reshape}(C_{gwc}) \in \mathbb{R}^{(G \cdot D) \times H \times W} \tag{2}$$

This operation explicitly encodes the disparity context into a unified dimension and allows our network to leverage the power of standard 2D convolutions to reason about complex 4D geometric representations without the overhead of 3D operations.

The reconstructed 3D cost volume $C_{3D}$ is coupled complex spatial information and disparity information. To efficiently extract required geometric representation, we propose the Bidirectional Geometry Aggregation (BGA) block with encoder-decoder architecture. The proposal of the BGA is based on the theoretical analysis in Section 3. We construct the BGA by repeatedly stacking Spatial Aggregation modules and Disparity Aggregation modules.

### 4.4 DISPARITY PREDICTION

After obtaining the aggregated cost volume, we apply the softmax operation to it to regress the disparity map $d_0$:

$$P = \text{Softmax}(\mathbf{C}_{agg}(d)), \tag{3}$$

$$\mathbf{d}_0 = \sum_{d=0}^{D_{max}/4-1} d \times P \tag{4}$$

where $D_{max}$ denotes the predefined maximum disparity. The disparity map $d_0$ is at $1/4$ resolution of input images. We utilize interpolation and learnable parameters respectively to upsample the disparity map $d_0$ to full resolution for supervision.

### 4.5 LOSS FUNCTION

We employ the smooth $L_1$ loss to supervise our network. The loss is defined as follow:

$$\mathcal{L} = \lambda_0 Smooth_{L_1}(\mathbf{d}_{init} - \mathbf{d}_{gt}) + \lambda_1 Smooth_{L_1}(\mathbf{d}_{final} - \mathbf{d}_{gt}) \tag{5}$$

where $d_{gt}$ is the ground truth of disparity and $\lambda_0 = 0.3, \lambda_1 = 1$.

Table 2: Results KITTI 2012 and KITTI 2015 online benchmarks. All results are taken from the official leaderboards or the original papers. The best and second best are marked with colors.

| Method | KITTI 2012 | | | | KITTI 2015 | | |
|---|---|---|---|---|---|---|---|
| | 3-noc | 3-all | 4-noc | 4-all | D1-bg | D1-fg | D1-all |
| DispNetC (Mayer et al., 2016b) | 4.11 | 4.65 | 2.77 | 3.20 | 4.32 | 4.41 | 4.34 |
| AANet (Xu & Zhang, 2020a) | 1.91 | 2.42 | 1.46 | 1.87 | 1.99 | 5.39 | 2.55 |
| DecNet (Yao et al., 2021) | - | - | - | - | 2.07 | 3.87 | 2.37 |
| CoEx (Bangunharcana et al., 2021) | 1.55 | 1.93 | 1.15 | 1.42 | 1.79 | 3.82 | 2.13 |
| DeepPruner-Fast (Xu et al., 2022) | - | - | - | - | 2.32 | 3.91 | 2.59 |
| HITNet (Tankovich et al., 2021) | 1.41 | 1.89 | 1.14 | 1.53 | 1.74 | 3.20 | 1.98 |
| Fast-ACVNet+ (Xu et al., 2023c) | 1.45 | 1.85 | 1.06 | 1.36 | 1.70 | 3.53 | 2.01 |
| Fast-ACVNet (Xu et al., 2023c) | 1.68 | 2.13 | 1.23 | 1.56 | 1.82 | 3.93 | 2.17 |
| MobileStereoNet-2D (Shamsafar et al., 2022) | - | - | - | - | 2.49 | 4.53 | 2.83 |
| MobileStereoNet-3D (Shamsafar et al., 2022) | - | - | - | - | 2.75 | 3.87 | 2.10 |
| BANet-2D (Xu et al., 2025) | 1.38 | 1.79 | 1.01 | 1.32 | 1.59 | 3.03 | 1.83 |
| BANet-3D (Xu et al., 2025) | 1.27 | 1.72 | 0.95 | 1.27 | 1.52 | 3.02 | 1.77 |
| DBStereo-S(Ours) | 1.81 | 2.29 | 1.24 | 1.62 | 1.92 | 3.73 | 2.24 |
| DBStereo-M(Ours) | 1.36 | 1.70 | 0.97 | 1.25 | 1.57 | 3.12 | 1.91 |
| DBStereo-L(Ours) | 1.25 | 1.60 | 0.91 | 1.14 | 1.50 | 2.98 | 1.77 |

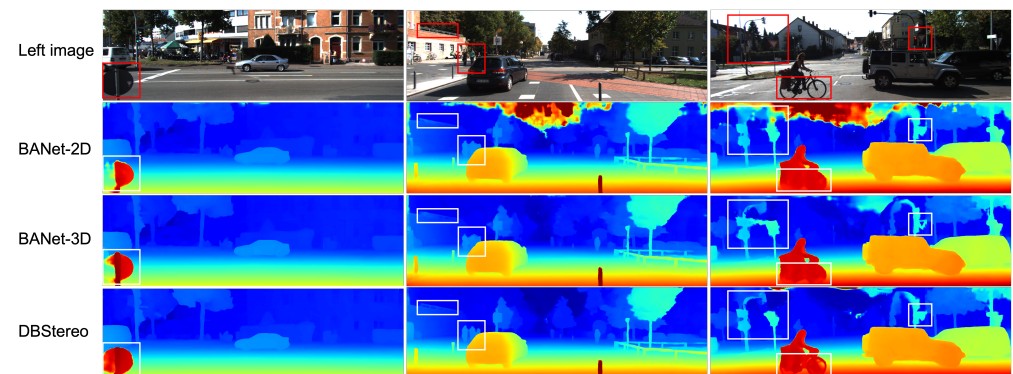

Figure 3: Qualitative results on KITTI test set. By employing the spatial-disparity decoupled aggregation, our method achieves outstanding performance in both texture-less and edge regions.

## 5 EXPERIMENTS

### 5.1 DATASETS AND EVALUATION METRICS

**Scene Flow** (Mayer et al., 2016a) is a large-scale synthetic stereo dataset containing 35,454 training and 4,370 testing stereo image pairs at 960×540 resolution. This dataset provides dense disparity map as ground truth. We utilize the End Point Error (EPE) and the D1 outlier as the evaluation metrics, where EPE is the average $l_1$ distance between the prediction and ground truth disparity. And D1 denotes the percentage of outliers with an absolute error greater than 1 pixels.

**KITTI** is a real-world dataset consisting of KITTI 2012 (Geiger et al., 2012) and KITTI 2015 (Menze & Geiger, 2015). KITTI 2012 provides 194 training pairs and 195 testing pairs, and KITTI 2015 provides 200 training pairs and 200 testing pairs. Both datasets provide sparse ground-truth disparities obtained with LiDAR.

### 5.2 IMPLEMENTION DETAILS

We have implemented our methods using PyTorch and conducted experiments on 8 NVIDIA RTX 3090 GPUs. We randomly crop images to 320 × 736 and use the same data augmentation as (Guo et al., 2025). We train our pretrained model on Scene Flow dataset for 90 epochs. For the KITTI

Table 3: Effectiveness of proposed modules on Scene Flow test set. SA denotes Spatial Aggregation, DA denotes Disparity Aggregation, 2D denotes 2D cost aggregation and 3D denotes 3D cost aggregation.

| Model | Cost volume | | Cost Aggregation | | | | EPE | D1 |
|---|---|---|---|---|---|---|---|---|
| | 3D | 4D | SA | DA | 2D | 3D | | |
| 3D Aggregation | | ✓ | | | | ✓ | 0.66 | 2.43 |
| 2D Aggregation | ✓ | | | | ✓ | | 0.68 | 2.57 |
| DBStereo(w/o DA) | | ✓ | ✓ | | | | 0.70 | 2.93 |
| DBStereo(w/o SA) | | ✓ | | ✓ | | | 0.73 | 3.19 |
| DBStereo(full model) | | ✓ | ✓ | ✓ | | | **0.63** | **2.36** |

online leaderboards, we fine-tuned the pre-trained model for 500 epochs using a mixed training set comprising KITTI 2012 and KITTI 2015 training datasets.

## 5.3 BENCHMARK DATASETS AND PERFORMANCE

We evaluated our DBStereo on the widely used Scene Flow benchmark. Additionally, to facilitate public comparison, we submitted our results to the official KITTI 2012 and KITTI 2015 online leaderboards.

**Scene Flow:** As shown in Table 1, we compare our proposed DBStereo variants with several state-of-the-art approaches on the Scene Flow dataset. Our DBStereo-L achieves the highest accuracy among all the published real-time methods and even surpass many high-performance iterative-based methods in both accuracy and inference time such as RAFT-Stereo and IGEV-Sterero, reducing the runtime by more than 85%. In addition, our DBStereo-S takes only 15ms while maintaining competitive performance. To more clearly demonstrate the advantage of DBStereo in the trade-off between efficiency and performance, we visualize the results from Table 1 in the bottom row of Figure 1, distinguishing between comparisons with aggregation-based methods and classical iterative-based methods. It can be observed that DBStereo achieves the optimal efficiency–accuracy curve.

**KITTI:** We fine-tuned the pre-trained model on the mixed dataset of KITTI 2012 and KITTI 2015 for best performance. As shown in the Table 2, our DBStereo-L achieved state-of-the-art performance for almost all metrics on KITTI 2012 and KITTI 2015 online leaderboards. Figure 3 shows qualitative results on KITTI 2012 and KITTI 2015 test sets, where our DBStereo significantly outperforms both 2D cost aggregation and 3D cost aggregation in the difficult scenarios.

## 5.4 ABLATION STUDY

We conducted comprehensive ablation studies to validate the contribution of each component in our framework. Due to the simplified training settings, the quantitative results of ablation experiments differ from the comparison results described above.

**Effectiveness of proposed modules** To demonstrate the effectiveness of the proposed Spatial-Disparity Decoupled Aggregation Paradigm compared to previous aggregation paradigms, we compared our DBStereo against corresponding variants that employ 3D convolutions to aggregate 4D cost volumes and 2D convolutions to aggregate 3D cost volumes. Specifically, for the variant corresponding to 3D aggregation, we removed the Disp2Channel operator and replaced both the Disparity Aggregation and Spatial Aggregation modules within the BGA block with a single 3D convolution. For the 2D aggregation variant, we substituted the 4D cost volume in DBStereo with a 3D correlation cost volume, while also removing the Disp2Channel operator and replacing the BGA block with standard 2D convolutions. The results presented in Table 3 indicate that DBStereo not only constructs a 4D cost volume enriched with geometric information but also achieves superior performance through the proposed decoupled 2D aggregation strategy. Furthermore, ablations involving the separate removal of Spatial Aggregation (SA) and Disparity Aggregation (DA) within DBStereo both led to significant performance degradation, underscoring the necessity of combining both aggregation mechanisms for optimal results.

**Sharper Unimodal Distribution from Disparity Aggregatio** In Sec 3, we posited that for an individual pixel, the true disparity value should be unique—that is, the disparity probability distribution should manifest as a sharp unimodal distribution. The disparity aggregation with global receptive field suppresses mismatches across the entire field of disparity, inherently introducing the inductive bias of this unimodal prior. To more intuitively demonstrate the advantage of our method, we visualize the disparity distributions corresponding to edge pixels, as shown in the upper right corner of Figure. 1. The visualized disparity probability distribution of our method is unimodal, whereas that of the original 3D convolution-based aggregation method exhibits a multimodal distribution.

**Runtime Analysis.** As quantitatively shown in Table 4, we provide a detailed comparison of inference latency across different model variants. All timing evaluations were conducted on a single RTX 3090 GPU with a batch size of 1. Although the computational demand naturally increases with model size and complexity, our approach maintains competitive inference speeds that satisfy real-time application requirements. Specifically, our DBStereo-S processes stereo image pairs at approximately 67 FPS, while the largest variant DBStereo-L still achieves a notable 21 FPS. These highlight the effectiveness of our design in optimizing the trade-off between real-time performance and prediction accuracy, demonstrating the scalability of our architecture and its practical usability in deployment.

Table 4: Runtime Analysis of DBStereo's different modules

| Module | Feature Extraction | Cost | Cost Aggregation | Disparity Regression | Total Time |
|---|---|---|---|---|---|
| DBStereo-S | 10.11 | 2.01 | 2.03 | 1.27 | 15.42 |
| DBStereo-M | 24.25 | 2.01 | 6.29 | 1.27 | 33.81 |
| DBStereo-L | 24.25 | 2.01 | 22.14 | 1.27 | 49.67 |

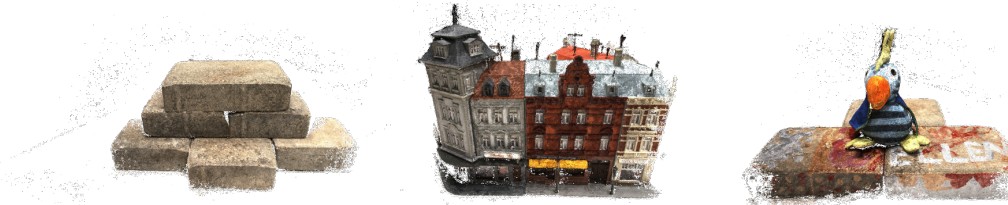

Figure 4: Visualization of results on DTU test set

## 6 EXTENSION TO MVS

We extend our DBStereo to multi-view stereo based on IterMVS. Following the training setting of IterMVS, we train our DBMVS on DTU dataset (Aanæs et al., 2016) for 32 epochs. As shown in Tab. 5, compared to IterMVS and its derivative IGEV-MVS, our approach achieves state-of-the-art performance in both accuracy and inference speed, which demonstrates the universality of our decoupled aggregation paradigm.

Table 5: Quantitative evaluation on DTU.

| Method | AbsRel ↓ | SqRel ↓ | RMSE ↓ | Runtime (ms) ↓ |
|---|---|---|---|---|
| IterMVS  (Wang et al., 2022) | 0.007 | 0.712 | 16.84 | 260 |
| IGEV-MVS  (Xu et al., 2023b) | 0.012 | 1.71 | 26.58 | 215 |
| DBMVS (Ours) | **0.006** | **0.522** | **13.30** | **62** |

## 7 CONCLUSION

In this paper, we provide a thorough analysis of the limitations of traditional aggregation paradigm methods, breaking the empirical approach of using dimension-matched convolutions for a high-dimensional cost volume. We propose the DBStereo which is based on pure 2D convolutions but achieve impressive performance both in accuracy and inferenc time. DBStereo is a simple yet strong baseline of our proposed decouple aggregation paradigm. We hope our research will provide some insightful directions for future community studies.

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

# A    APPENDIX

## A.1    THE USE OF LARGE LANGUAGE MODELS

The authors confirm their full accountability for the scholarly validity and originality of this manuscript. We attest that artificial intelligence was in no way used to generate or falsify research data. The only application of Large Language Models was to aid in wording and phrasing, with the goal of improving the prose's idiomatic flow and making the presentation more accessible to an international academic audience. The final responsibility for the intellectual content and its expression remains entirely with the authors.

