# OpenReview forum: "Decoupling Bidirectional Geometric Representations of 4D cost volume via 2D convolution"
_ICLR.cc/2026/Conference — ICLR 2026 Conference Withdrawn Submission_

### Official Review · Reviewer_HRyD · 2025-10-28

**Soundness:** 2
**Presentation:** 3
**Contribution:** 2
**Rating:** 6
**Confidence:** 4

**Summary:**

1. A decoupled bidirectional aggregation strategy that treats spatial and disparity representations separately, with a proposed spatial aggregation module to capture local spatial context for each disparity level and a disparity aggregation module to possess a global receptive field.
2. The proposed stereo matching baseline replaces computationally expensive 3D convolutions with lightweight 2D convolutions for processing the 4D cost volume, reducing computational complexity while effectively mitigating the multi-peak disparity prediction issue commonly caused by 3D convolutions.

**Strengths:**

1. Demonstrates real-time performance with significantly reduced computational cost by using a pure 2D convolution framework.
2. Introduces the Bidirectional Geometry Aggregation (BGA) module, enabling clearer unimodal disparity distributions and sharper object boundaries.
3. Presents a simple yet strong baseline that is highly deployable and extensible to multi-view stereo tasks.

**Weaknesses:**

1. The discussion and explanation regarding multi-modal vs. unimodal disparity distributions are rather general. Apart from the illustrative example in Figure 1, the paper lacks quantitative analysis or empirical evidence to substantiate the claim that the proposed simple disparity aggregation effectively promotes unimodal disparity predictions.

2. The paper does not include experiments that replace 3D convolutions with the proposed decoupled 2D convolutions within existing cost-volume-based stereo matching frameworks, which would be valuable for demonstrating the generality and robustness of the proposed approach.

3. The comparison on the KITTI datasets (Table 2) is not sufficiently comprehensive, as it only includes relatively lightweight stereo matching models, without evaluation against a broader range of state-of-the-art or large-scale methods.

**Questions:**

1. Generality of the proposed approach:
   Can the authors clarify whether the proposed decoupled 2D convolution paradigm is generally applicable to other state-of-the-art cost-volume–based stereo matching networks (e.g., IGEV-Stereo or RAFT-Stereo)? Specifically, if the 3D convolution modules in these methods were replaced with the proposed 2D decoupled aggregation, would the model still achieve improved accuracy and real-time performance? Demonstrating such generality would strengthen the paper’s contribution.

2. Empirical evidence for unimodal disparity distributions:
   The paper claims that the disparity aggregation effectively promotes a unimodal disparity distribution, leading to clearer object boundaries. Could the authors provide more convincing qualitative and quantitative evidence, (e.g., statistical analysis of disparity distributions or additional visual comparison) to better support this claim?

---

### Official Review · Reviewer_wvpG · 2025-10-31

**Soundness:** 2
**Presentation:** 3
**Contribution:** 3
**Rating:** 4
**Confidence:** 4

**Summary:**

The paper proposes a decoupled aggregation framework for stereo/MVS in which spatial aggregation and disparity aggregation are handled by two dedicated branches. Instead of heavy 3D convolutions on the cost volume, the method uses a spatial module to capture multi-scale image context and a disparity module to model long-range relationships along the disparity axis. This design aims to mitigate the limited receptive field and overfitting issues of 3D convs, while reducing parameters, memory, and latency.

**Strengths:**

1.The paper clearly analyzes two drawbacks of 3D convolutions in learning-based MVS method, and proposes targeted solutions. The reasoning is sound and easy to follow.
2.The decoupling idea (spatial vs. disparity) is simple and modular. It can be plugged into existing pipelines and should be compatible with common backbones and cost-volume designs.
3.The writing is clear and well organized; the method pipeline is easy to understand and implement.

**Weaknesses:**

1.The author claim improved efficiency, but Table 3 does not report the key metrics. Please add, for example, a full comparison between 3D aggregation and your 2D aggregation on the same hardware in terms of FPS, GPU memory, parameters (M).
2.Table 5 better shows your advantages (please state clearly whether it is the small or large model), but the dataset coverage is too narrow (only DTU). It would be better to add results on more datasets (e.g., KITTI, ETH3D, Tanks&Temples if applicable) and provide more qualitative comparisons in the future main text or appendix.
3.Generalization and “3D-conv overfitting” claim lack evidence.
The author state that “this coupled learning approach is susceptible to overfitting due to noise in the training data”, but there is no direct experimental support. Please add cross-dataset and zero-shot tests, for example: Train on KITTI and test zero-shot on DrivingStereo or Train on SceneFlow and test on KITTI (zero-shot). This will make the “better generalization” claim concrete.

**Questions:**

1.What is the reason for the baseline mismatch between Table 1 and Table 2? In particular, why isn’t methods like IINet included in Table 2, and why are runtime numbers omitted there? Since BANet-3D appears stronger than DBStereo-M in Table 2 and their runtimes are similar in Table 1, could you add the missing results to Table 2?
2.The author state that the coupled learning approach is susceptible to overfitting due to noise in the training data. Why does training-data noise lead to overfitting in this setting?
3.In the “Extension to MVS” section, why compare only with iterative optimization–based and not with cost-volume–based methods?

---

### Official Review · Reviewer_9m1e · 2025-11-01

**Soundness:** 2
**Presentation:** 2
**Contribution:** 2
**Rating:** 2
**Confidence:** 3

**Summary:**

The manuscript proposed DBStereo, which decouples the traditional 3D aggregation into spatial aggregation and disparity aggregation, and outperforms some existing methods in terms of inference time and speed.

**Strengths:**

1. The design is generalized to multi-view stereo and outperforms IterMVS and IGEV-MVS.

**Weaknesses:**

1. The decomposition of 3D convolution has been used in stereo matching. For example, [1][2].
2. Some recent works are missing in Tab 2. For example, [3][4][5][6].
3. Could the disparity aggregation module be generalized to wider range of disparities during test? If yes, would it be possible to provide experiments, such as on Middlebury or ETH3D?

[1] Separable Convolutions for Optimizing 3D Stereo Networks
[2] FoundationStereo: Zero-Shot Stereo Matching
[3] BridgeDepth: Bridging Monocular and Stereo Reasoning with Latent Alignment
[4] DEFOM-Stereo: Depth Foundation Model Based Stereo Matching
[5] IGEV++: Iterative Multi-range Geometry Encoding Volumes for Stereo Matching
[6] DS-Stereo: Deep-Shallow Information Interaction for Stereo Matching

**Questions:**

1. In Figure 1, what does reg_disp mean? Would it be possible to provide more examples to demonstrate the DBStereo could predict unimodal distribution, especially at object boundaries?
2. In Figure 2, what does MSCA mean?
3. Would it be possible to provide detailed configurations for  DBStereo-S,  DBStereo-M, and DBStereo-L, respectively?

---

### Official Review · Reviewer_eV1t · 2025-11-01

**Soundness:** 2
**Presentation:** 3
**Contribution:** 2
**Rating:** 4
**Confidence:** 4

**Summary:**

In this paper, the authors aim to design a lightweight cost aggregation network for real-time, high-quality stereo reconstruction. The paper analyzes the limitations of traditional 3D CNN–based aggregation methods and discusses the inherent inductive bias of the 4D cost volume. Based on these observations, a specially designed disparity–spatial 2D aggregation network is proposed to regularize the cost volume in a lightweight manner. The authors compare both performance and efficiency to demonstrate the effectiveness of their design.

**Strengths:**

1. The paper is mostly well-written and easy to follow.
2. The authors validate the effectiveness of their method across multiple datasets and conduct comprehensive ablation studies to demonstrate the effectiveness of each module.

**Weaknesses:**

- Claimed decoupling unconvincing: The authors claim that 3D CNNs overly couple spatial and disparity dimensions, and address this by reshaping the cost volume from (B, C, D, H, W) to (B, C×D, H, W) for 2D CNN processing. However, this reshaping does not actually decouple the two dimensions. Each spatial position still aggregates information from all disparity channels through 2D convolution, meaning spatial and disparity features remain mixed. The authors should clarify how their design effectively reduces such coupling. A more effective way to achieve decoupling is to merge the disparity dimension into the batch dimension and process each disparity level independently with 2D CNNs, as done in previous work such as [1].
- Limited performance gain: In Table 3, the ablation study shows that the full model achieves only a modest improvement over the 3D CNN baseline; the performance improvement brought by the proposed design appears limited.
- Limited novelty: Separating spatial and disparity aggregation in cost volumes has been widely studied. Although the authors report good performance, the contribution is limited.
- Missing related work: Recent work LightStereo [2] designs lightweight aggregation networks based on 2D CNNs. The authors should compare both performance and efficiency with it.
- Minor issues: Table 2 should include a runtime comparison similar to Table 1; the reference around L516–553 appears to be duplicated.

**Questions:**

1. As mentioned above, the 2D convolution for spatial aggregation still effectively aggregates information across the entire disparity dimension. Compared to standard 3D convolution, this reshaping does not truly decouple spatial and disparity features; in fact, folding the disparity into the channel dimension may even increase the disparity receptive field (which may be the main source of the observed performance gain). The authors need to provide a clear explanation for this design choice and its effect.
2. It would be better to include comparisons of runtime and parameter count in Table 2. The performance gain of the proposed model over the baseline (4D cost + 3D CNN) is limited, and improvements in efficiency could help justify the proposed design.

---

### Note · Authors · 2025-11-12

I have read and agree with the venue's withdrawal policy on behalf of myself and my co-authors.